# RNA-Sequencing of Tumor-Educated Platelets, a Novel Biomarker for Blood-Based Sarcoma Diagnostics

**DOI:** 10.3390/cancers12061372

**Published:** 2020-05-27

**Authors:** Kimberley M. Heinhuis, Sjors G. J. G. In ’t Veld, Govert Dwarshuis, Daan van den Broek, Nik Sol, Myron G. Best, Frits van Coevorden, Rick L. Haas, Jos H. Beijnen, Winan J. van Houdt, Tom Würdinger, Neeltje Steeghs

**Affiliations:** 1Department of Medical Oncology and Clinical Pharmacology, The Netherlands Cancer Institute, Plesmanlaan 121, 1066 CX Amsterdam, The Netherlands; k.heinhuis@nki.nl; 2Department of Neurosurgery, Amsterdam University Medical Center, Vrije Universiteit Medical Center, Cancer Center Amsterdam, De Boelelaan 1117, 1081 HV Amsterdam, The Netherlandsgovert.dwarshuis@hotmail.com (G.D.); m.g.best@amsterdamumc.nl (M.G.B.); 3Brain Tumor Center Amsterdam, Amsterdam University Medical Center, Vrije Universiteit Medical Center, Cancer Center Amsterdam, De Boelelaan 1117, 1081 HV Amsterdam, The Netherlands; ni.sol@amsterdamumc.nl; 4Department of Laboratory Medicine, The Netherlands Cancer Institute, Plesmanlaan 121, 1066 CX Amsterdam, The Netherlands; da.vd.broek@nki.nl; 5Department of Neurology, Amsterdam University Medical Center, Vrije Universiteit Medical Center, Cancer Center Amsterdam, De Boelelaan 1117, 1081 HV Amsterdam, The Netherlands; 6Department of Pathology, Amsterdam University Medical Center, Vrije Universiteit Medical Center, Cancer Center Amsterdam, De Boelelaan 1117, 1081 HV Amsterdam, The Netherlands; 7Department of Surgical Oncology, The Netherlands Cancer Institute, Plesmanlaan 121, 1066 CX Amsterdam, The Netherlands; coevor@me.com (F.v.C.); w.v.houdt@nki.nl (W.J.v.H.); 8Department of Radiotherapy, The Netherlands Cancer Institute, Plesmanlaan 121, 1066 CX Amsterdam, The Netherlands; r.haas@nki.nl; 9Department of Pharmacy & Pharmacology, The Netherlands Cancer Institute, Plesmanlaan 121, 1066 CX Amsterdam, The Netherlands; j.beijnen@nki.nl; 10Department of Pharmaceutical Sciences, Utrecht University, Universiteitsweg 99, 3584 CG Utrecht, The Netherlands

**Keywords:** liquid biopsy, sarcoma, messenger RNA, blood platelets

## Abstract

Sarcoma is a heterogeneous group of rare malignancies arising from mesenchymal tissues. Recurrence rates are high and methods for early detection by blood-based biomarkers do not exist. Hence, development of blood-based liquid biopsies as disease recurrence monitoring biomarkers would be an important step forward. Recently, it has been shown that tumor-educated platelets (TEPs) harbor specific spliced ribonucleic acid(RNA)-profiles. These RNA-repertoires are potentially applicable for cancer diagnostics. We aim to evaluate the potential of TEPs for blood-based diagnostics of sarcoma patients. Fifty-seven sarcoma patients (active disease), 38 former sarcoma patients (cancer free for ≥3 years) and 65 healthy donors were included. RNA was isolated from platelets and sequenced. Quantified read counts were processed with self-learning particle-swarm optimization-enhanced thromboSeq analysis and subjected to analysis of variance (ANOVA) statistics. Highly correlating spliced platelet messenger RNAs (mRNAs) of sarcoma patients were compared to controls (former sarcoma + healthy donors) to identify a quantitative sarcoma-specific signature measure, the TEP-score. ANOVA analysis identified distinctive platelet RNA expression patterns of 2647 genes (false discovery rate <0.05) in sarcoma patients as compared to controls. The self-learning algorithm reached a diagnostic accuracy of 87% (validation set only; *n* = 53 samples, area under the curve (AUC): 0.93, 95% confidence interval (CI): 0.86–1). Our data indicates that TEP RNA-based liquid biopsies may enable for sarcoma diagnostics.

## 1. Introduction

Sarcoma is a malignancy arising from the connective tissue or bones, with an incidence of less than 1% in adults and 20% in children [1,2]. Sarcoma is a heterogeneous group of cancers with over 70 different histologic subtypes, and can arise throughout the whole body [1,3]. Due to a lack of tumor specific symptoms there is often a delay in diagnosis [2]. Approximately 25% of all sarcoma patients will develop distant metastases, rising up to 40–50% for sarcoma with high risk features [4,5]. Therefore, currently, most sarcoma patients are being followed for up to 10 years in order to detect recurrences. Early detection of low stage disease in the primary setting is associated with a better outcome, caused by improved resectability of the tumors, but lead-time bias should be taken into account [1,6]. Accurate diagnostic tools to achieve a correct and early diagnosis are important to potentially improve survival of sarcoma patients. Whereas in many cancer types blood-based markers have been developed to screen for disease recurrence, such as carcinoembryonic antigen (CEA) for colorectal cancer and prostate specific antigen (PSA) for prostate cancer [7,8], there are no clinically implemented tumor markers for sarcoma. Blood-based biomarkers have several advantages, including the low cost of screening, the relatively low patient burden compared to imaging and the lack of radiation exposure in the screening program. Also, follow-up with a low-invasive diagnostic tool instead of the more invasive screening program, which is currently applied, could improve the quality of life of sarcoma patients and an early detection method increases the chance of a better survival [6].

The applicability of several biomarkers in blood-based diagnostics are currently investigated, but most approaches show a lack in specificity in diagnosing the primary tumor [9,10]. A technique which can potentially overcome this problem of specificity is the blood-based diagnostic tool which analyses tumor-educated platelets (TEPs) [9,11,12,13]. Blood platelets are widely known for their role in hemostasis. However, recent research has revealed their contribution to the progression and metastasis of cancer [14], including in sarcomas [15,16,17,18]. Blood platelets contain messenger ribonucleic acids (mRNAs) which may undergo specific splice events in response to external stimuli, potentially induced by the primary tumor [9]. Such external queues may result in tumor-specific RNA-profiles that may be employed for blood-based cancer diagnostics and monitoring of tumor recurrence [9,11]. Previously, a self-learning thromboSeq algorithm has been employed to detect several forms of cancer, such as non-small cell lung cancer (NSCLC), colorectal cancer, glioblastoma, breast cancer, pancreatic cancer, and hepatobiliary cancer [9,11,19,20]. The aim of this study was to investigate whether TEPs could be used as a blood-based diagnostic tool in the detection of sarcoma patients.

## 2. Results

We included 57 patients with active sarcoma disease, and 103 controls of which 38 were former sarcoma patients (at least three years free of cancer and anti-cancer treatment) and 65 were individuals reported to have no cancer (healthy donors).

To prevent potential confounding effects of the variables age and gender, both series were matched [21]. The demographic characteristics are summarized in Table 1 and the distribution of the histologic subtypes of the sarcoma series are provided in Table 2. The majority of sarcoma patients had metastatic disease (68%) and the most prevalent histological subtypes included were liposarcoma, gastrointestinal stromal tumor, and leiomyosarcoma. We were unable to calculate a correlation between disease-stage and TEP-score because of the limited number of stage II samples (*n* = 2).

In total, following filtering and quality steps (Appendix A), 3,799 RNAs with sufficient read coverage were identified in the platelet profiles. To circumvent potential cell-free DNA contamination, the thromboSeq pipeline only includes spliced RNA reads (reads from exon-to-exon or intron-spanning, also termed “spliced junctions”) for downstream analyses. We first compared all sarcoma patients (*n* = 57) to all controls (*n* = 103) by analysis of variance (ANOVA) analysis, resulting in 2647 RNAs with differential expressed spliced junctions (false discovery rate (FDR) < 0.05; Appendix A). Unsupervised clustering of particle-swarm optimization (PSO)-enhanced FDR selection (2537 RNAs, FDR < 0.033) resulted in clear separation of sarcoma patients and controls (Figure 1a, *p* < 1.6 × 10^−6^). Hence, we concluded that TEP RNA is significantly altered in patients with sarcoma as compared to healthy donors and patients with no active disease, and resulted in a differentially expressed spliced junction RNA signature.

Subsequently, we developed a self-learning classification algorithm that enables for independent diagnosis of patients with sarcoma. For this, we separated the complete dataset into training, evaluation, and validation series. Here, the training series is employed for biomarker panel selection and training of a self-learning support vector machine (SVM)-algorithm. Subsequently, the performance of this compiled biomarker panel and SVM-algorithm was evaluated in the evaluation series after which new instructions regarding biomarker panel selection thresholds were provided to the training series using particle-swarm optimization. This process was performed 1000 times in order to improve classification accuracy in the evaluation series. Once satisfied, the algorithm was locked and an independent set of validation samples was classified. The training series consisted of 21 sarcoma samples and 34 controls, the evaluation series consisted of 19 sarcoma samples and 33 controls, the validation series consists of 17 sarcoma samples and 36 controls (Table 1). The optimization process resulted in an optimum detection accuracy of 90% in the evaluation series, applying the default cutoff of the TEP-score of 0.5 (number (*n*) = 52 samples, area under the curve (AUC): 0.94, 95% confidence interval (CI) 0.87–1, Figure 1b, red line), and a detection accuracy of 88% in the validation series (*n* = 53 samples, AUC: 0.93, 95% CI: 0.86–1, *p* < 0.001, Figure 1b, blue line). Post-hoc evaluation of the training series using a leave-one-out cross validation (LOOCV) approach resulted in similar detection rates (accuracy: 85%, AUC: 0.92, 95% CI 0.88–1, Figure 1b, grey line). In order to assess uniqueness of sarcoma signature, we performed a Venn diagram analysis of this signature as compared to the signature identified in Best et al. Cancer Cell 2017 [11] (NSCLC versus non-cancer controls) and Best et al. Nature Protocols 2019 [20] (lower-grade glioma (LGG) versus healthy donors). We observed in total an overlay of 66 (2%) spliced RNAs between all three signatures, whereas the majority appears to be uniquely present in any of the signatures (for LGG: 1168/1711 (68%); for NSCLC: 533/1000 (53%); for sarcoma: 472/824 (57%), Appendix A). Hence, we conclude that a unique sarcoma signature can be selected from TEP RNA profiles. A variant of our sarcoma classifier trained on the subset of 472 sarcoma specific transcripts as visualized in Appendix A, did not outperform the original swarm-enhanced biomarker panel of 884 transcripts (Appendix A).

To provide better insight in the classification of patients with sarcoma versus controls, we determined the distribution of the TEP-scores of the different samples in the training, evaluation and validation series (Figure 2). The TEP-score ranges from zero to one and represents the algorithms’ measure of the expression of the sarcoma profile in a particular sample. A TEP-score of 0.5 was used as a cut-off value for an adequate differentiation between the diagnosis sarcoma and the diagnosis healthy. The cut-off value was based on the most accurate fit with a high sensitivity, specificity and accuracy. Because the aim of our study is to evaluate the potential of TEPs in sarcoma diagnostics, the exact cut-off of 0.5 was selected to obtain high specificity to avoid an excess of false-positive samples. The default cutoff of the SVM algorithm (0.5) resulted in a specificity of 88%, sensitivity of 81% and accuracy of 85% for the training series, a specificity of 94%, sensitivity of 84% and accuracy of 80% for the evaluation series, and a specificity of 86%, sensitivity of 88%, and accuracy of 87% for the validation series.

The histologic subtypes of the nine sarcoma patients who were wrongly classified as control (in either the training, evaluation or validation series) were: Four patients with locally-advanced or metastasized dedifferentiated liposarcoma; two patients with metastasized leiomyosarcoma; two patients with small lesions of gastrointestinal stromal tumor; and one patient with localized myxoid liposarcoma. Four of these patients had stable disease. The tumor size and tumor stage of the outliers was not different from the correctly classified sarcoma patients. The histologic subtypes of the eleven controls who were wrongly classified as sarcoma (in either the training, evaluation and validation series) were: One healthy donor; three patients with fibrosarcoma; three patients with myxoid liposarcoma; two patients with gastrointestinal stromal tumor; one patient with angiosarcoma; one patient with an alveolar soft part sarcoma in their medical history. None of the wrongly classified former sarcoma patients had a recurrence at their next follow-up visit with a median follow-up time of 24.7 months after the blood draw.

## 3. Discussion

The goal of this proof-of-concept study was to investigate whether TEPs can be employed as a blood-based biomarker tool for sarcoma patients. We showed that TEP RNA is significantly altered in the presence of sarcoma as opposed to former sarcoma patients and healthy donors. With the developed TEP-score, most sarcoma patients could be identified and distinguished from control samples. Among the false negative sarcoma patient samples, there was no specific histologic sarcoma subtype more abundant. We showed that thromboSeq might be a promising technique in sarcoma diagnostics, however its potential in monitoring of tumor recurrence still needs to be explored. The follow-up of sarcoma patients currently consists of physical examination and radiologic imaging every few months. Additional blood sampling for the TEP RNA-based analysis could be obtained during the standard follow-up visits to investigate if recurrences may be objectified in an earlier phase compared with radiologic imaging alone. Whether this screening method can be used for all sarcoma types remains to be investigated. Creating a single TEP-score for sarcoma might be challenging due to the heterogeneity of the sarcoma population. However, there are currently no clinical relevant blood-based markers available for sarcoma patients, and therefore development of a new tool is warranted.

The various techniques for liquid biopsies which are currently under investigation for their application in sarcoma can be divided in four groups of biosources: Circulating tumor cells (CTCs), circulating cell-free nucleic acids (ccfNAs), exosomes, and metabolites [15]. The advantage of CTCs is that multiple components can be investigated like DNA, RNA, and proteins. Unfortunately, sarcoma patients only have a limited amount of CTCs in their bloodstream, and therefore a larger sample volume would need to be collected. Moreover, it seems hard to determine relevant aberrations in CTCs of sarcoma patients [10]. ccfNAs are considered to have a higher sensitivity than CTCs [22]. Exosomes can be measured in all kind of body fluids and are related to angiogenesis and metastasis, which makes them a promising biosource to predict tumor progression and metastasis. However, most exosomes lack tumor-specific markers [10]. Another biosource is composed of metabolites, which are considered representative of the tumor phenotype. Even though they may offer more detail on potential targets for therapy, these metabolites may also be quite sensitive to physiological and chemical changes of the environment [15]. Here, with the PSO-enhanced thromboSeq an AUC of 0.93 in a 53-samples validation series was achieved, which makes the TEP-score a potentially sensitive and specific tool.

TEPs have been tested in other cancer types as well, and the accuracy for sarcoma is comparable [9,11,19,20,21]. There are several limitations to this proof-of-concept study. Although we included sufficient sarcoma samples for an initial classification, and a separation between sarcoma and controls was observed, in the current analysis former sarcoma patients were pooled with the asymptomatic controls to have a better age- and gender-status-matched series. Despite the fact that former sarcoma patients classified more as control than as sarcoma (Figure 2), this creates a potential bias. Another limitation of this study is that we cannot yet differentiate between the different histologic subtypes and different stages of diseases of sarcoma. To improve this ability of the algorithm, we need to include more patients per histologic subtype and with lower stage of disease in a prospective study. Finally, analysis of the biological function of the altered spliced TEP transcripts is required to further understand the role of platelets in patients with sarcoma.

## 4. Materials and Methods

### 4.1. Inclusion of Patients

Blood samples were collected from sarcoma patients with active tumor load before or during anti-cancer treatment (sarcoma series). All samples were processed according the same standardized protocol of Best et al. [20]. Patients were ≥18 years, and written informed consent had to be obtained. All histologic subtypes of sarcoma were eligible. Two samples were collected at different time points in patient treatment in two cases in the sarcoma validation series. Samples of former sarcoma patients (defined as at least three years free of cancer and anti-cancer treatment) and healthy donors were also collected. Healthy donors reported to be without any type of cancer, currently or in the past. Former sarcoma patients and healthy donors were pooled in a control series, which was age- and gender-matched to the sarcoma patient sample series. The samples and associated clinical data of all individuals was collected and stored with a retraceable code, and fully anonymized.

This study was approved by the institutional review board of the Netherlands Cancer Institute under number CFMPB420.

### 4.2. Particle-Swarm Optimization-Enhanced ThromboSeq Analysis

For the RNA extraction, sequencing and interpretation we used the standardized protocol of Best et al. [20]. In the first step the blood platelets were isolated from whole blood samples and the RNA was extracted and sequenced with SMARTer-based complementary deoxyribonucleic acid (cDNA) synthesis and amplification and Illumina TruSeq cDNA labeling and sequencing on the Illumina platform. All steps are quality-controlled by Bioanalyzer analysis. Secondly, the blood platelet RNA-sequencing data was processed and used for the development and validation of the PSO-enhanced classification algorithm. To circumvent potential cell-free DNA contamination, only intron-spanning spliced RNA reads were selected for analysis. ANOVA was used to determine the difference in the level of spliced RNAs between sarcoma patients and control samples. The panel resulting in the most optimal separation of the groups after unsupervised clustering was visualized in the PSO-enhanced heatmap according to the default settings as published before (12 iterations and 200 particles, 1200 particles in total). For PSO-enhanced algorithm development, we applied the predefined settings; libsize correlation between −0.1 and 1.0, FDR between 0.00001 and 1.0, correlated transcripts between 0.5 and 1.0 and ranked transcripts between 200 and all 3,799 detected transcripts. We selected the particle (algorithm settings) showing the best performance on the evaluation set after creating 100 particles during 10 iterations (1000 particles in total). The particle resulting into the highest AUC on the evaluation series was applied post-hoc on the training set (LOOCV) and on the validation series. A FDR-threshold of <0.05 was stated as statistically significant [9,11]. A sarcoma TEP-score was generated as a measure for the probability of a sample belonging to the sarcoma cohort.

## 5. Conclusions

TEP-based liquid biopsies can potentially be used as a blood-based diagnostic tool for sarcoma patients. The PSO-enhanced thromboSeq analysis of TEPs is a highly sensitive and specific tool for the detection of sarcoma. Algorithm optimization by including more prospectively collected samples into the development process is likely to improve the reproducibility of the test. Hence, a prospective study is warranted.

## 6. Patents

M.G.B. and T.W. are inventors on relevant patent applications. 

## Figures and Tables

**Figure 1 cancers-12-01372-f001:**
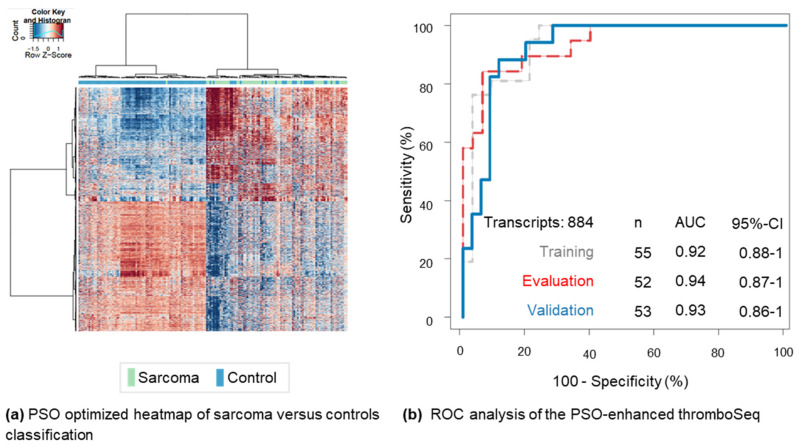
TEP thromboSeq analysis of sarcoma patients versus controls. (**a**) Particle-swarm optimization (PSO) optimized heatmap of sarcoma patients versus controls. Out of 3799 spliced transcripts detected in the blood platelets, 2647 transcripts were differentially expressed spliced junctions (false discovery rate < 0.05) and 2537 transcripts were used for clustering analysis (false discovery rate < 0.033). Unsupervised hierarchical clustering of differentially expressed spliced junction ribonucleic acid (RNA) transcripts between controls (blue, number (*n*) = 103) and sarcoma patients (green, *n* = 57). The columns indicate the different patient samples. The rows indicate the differentially expressed spliced junction RNA transcripts. The color intensity represents the Z score-transformed RNA expression value. (**b**) Receiver operating characteristics (ROC) analysis of the PSO-enhanced thromboSeq classifications using controls, including healthy donors and former sarcoma patients, and patients with sarcoma. The red dashed line indicates the ROC of the evaluation series (*n* = 52), classified by the self-learning support vector machine (SVM) algorithm developed with 884 transcripts and spliced RNA levels derived from the training series (Appendix A). The grey dashed line indicates the post-hoc assessment of the 884 transcripts by leave-one-out-cross-validation (LOOCV) on the training series (*n* = 55). The blue line indicates the ROC of the validation series (*n* = 53). As an internal control experiment, shuffled class labels (*n* = 1000) resulted a median area under the curve (AUC) in the validation series of 0.47 and an interquartile range (IQR) of 0.36. Shuffled samples in training (1000 iterations) resulted in a median AUC in the validation series of 0.92 with an IQR of 0.05.

**Figure 2 cancers-12-01372-f002:**
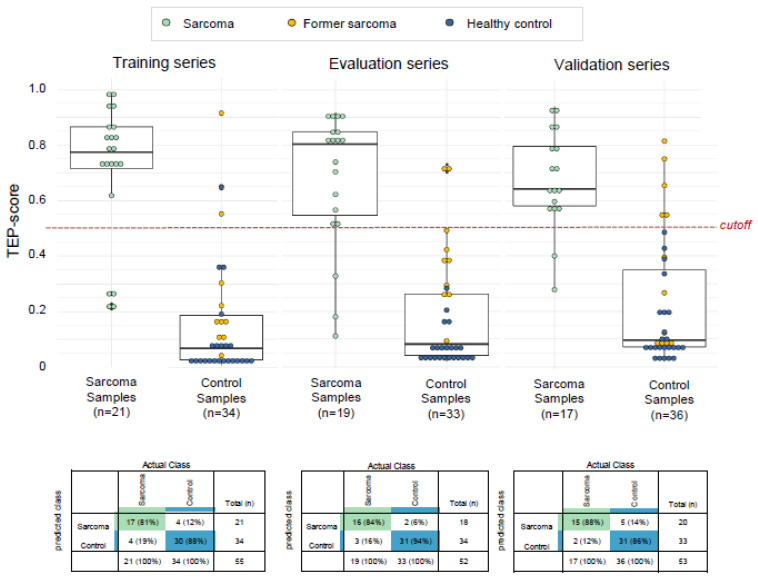
Distribution of the tumor-educated platelets (TEP)-score for the training, evaluation and validation series. The distribution of the TEP-score for the training, evaluation and validation series for both discriminative groups, patients with sarcoma and controls. Samples are colored by their cancer status. In green, patients with present sarcoma tumor load, in blue, healthy donors without any history of cancer, in yellow, samples obtained from former sarcoma patients who are currently monitored for disease recurrence. Below are per series shown 2 × 2 cross-tables, indicated are sample numbers and detection rates in percentages.

**Table 1 cancers-12-01372-t001:** Demographic characteristics. The demographic characteristics of the sarcoma and controls (former sarcoma patients and healthy donors) are shown, and their distribution between the training, evaluation, and validation series.

Characteristics	Training		Evaluation		Validation	
Sarcoma cohort N	21		19		17	
Median age (IQR) in years	56 (19)		60 (18)		60 (19)	
F/M %	33/67		74/26		35/65	
Localized N (%)	5 (24%)		7 (37%)		5 (29%)	
Metastasized N (%)	16 (76%)		12 (63%)		12 (71%)	
Controls cohort N	34		33		36	
Median age (IQR) in years	61 (20.5)		59 (18)		54 (26.5)	
F/M %	55/45		85/15		58/42	
Former sarcoma /healthy donors N	11	23	10	23	12	24
Median age (IQR) in years	72 (13.5)	57 (23)	59.5 (17)	59 (17)	64 (11.5)	48 (22.5)
F/M %	55/45	100/0	50/50	100/0	58/42	100/0

F = female, IQR = interquartile range, M = male, N = number of patients.

**Table 2 cancers-12-01372-t002:** Overview of histologic subtypes of the included sarcoma patients. Distribution of the different histologic subtypes of sarcoma patients between the training, evaluation and validation series.

Histologic Subtype	Training	Evaluation	Validation
Dedifferentiated liposarcoma	3	1	2
Myxoid liposarcoma	3	2	1
Pleomorphic liposarcoma	1	1	0
Leiomyosarcoma	3	3	5
Gastrointestinal Stromal Tumor	5	5	6
Myxofibrosarcoma	1	2	0
Undifferentiated pleomorphic sarcoma	1	1	1
Others	1	1	0
Angiosarcoma	1	0	1
Ewing sarcoma	0	0	1
Extraskeletal chondrosarcoma	1	1	0
Malignant peripheral nerve sheath tumor			
Synovial sarcoma	1	2	0
Total	21	19	17

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
