# Peer review of "RNA-Sequencing of Tumor-Educated Platelets, a Novel Biomarker for Blood-Based Sarcoma Diagnostics"

_cancers, 2020, doi:10.3390/cancers12061372_

Round 1

Reviewer 1 Report

In this well-written manuscript, the authors describe the potential use of tumor-educated platelets (TEPs) for blood-based diagnostics of sarcoma tumor.

The authors shown that TEP RNA is significantly altered in patients with sarcoma as compared to healthy individuals and patients with no active disease. The results are obtained through the optimization-enhanced thromboSeq analysis; the AUC of 0.93 in validation was achieved.

The authors conclude that a unique sarcoma signature can be selected from TEP RNA profiles making the TEP-score a potentially sensitive and specific tool. The results of this study indicate that blood platelets are a potential a platform for blood-based cancer diagnostics, using the equivalent of one drop of blood.

The findings are described in a convincing manner However, the following points might be addressed by the authors to further improve the significance and impact of this report on the field:

1 Regarding the heterogeneity of sarcomas, the authors should report information that summarizes the clinical characteristics of patients included in the study (age, sex, primary site, size, metastatic site) and how the selection of the cases occurred. Table 2 gives an overview of histologic subtypes of the included sarcoma patients. The authors should expand information regarding the origin of the tumor, if they derive from soft tissues or bone. It seems that in the series of fifty-seven patients there are one patient with sarcoma arising from bone (Ewing's sarcoma); how the authors justify the lack of bone sarcoma?

2 Regarding patients with active sarcoma, the authors should give precise information about the time of collection. It would be interesting to know if the blood samples for TEPs analysis were collected at the time of diagnosis, pre- or post-opratively or during follw-up. Is there homogeneity when collecting the blood sample in patients with active sarcoma? Samples for training and validation evaluation were collected and processed similarly?

3 During the platelet-isolation procedure it seems possible the residual circulating free DNA contamination potentially introduced causes reduced numbers of platelet-RNA sequencing reads. The authors should explain what kind of measures have been implemented to avoid this inconvenience and have false results

Authors should follow the style of a structured abstract, which is based on the IMRAD structure of a paper.

Author Response

Comments of reviewer #1: 1. Regarding the heterogeneity of sarcomas, the authors should report information that summarizes the clinical characteristics of patients included in the study (age, sex, primary site, size, metastatic site) and how the selection of the cases occurred. Table 2 gives an overview of histologic subtypes of the included sarcoma patients. The authors should expand information regarding the origin of the tumor, if they derive from soft tissues or bone. It seems that in the series of fifty-seven patients there are one patient with sarcoma arising from bone (Ewing's sarcoma); how the authors justify the lack of bone sarcoma?

Response: The reviewer highlights an important point about the heterogeneity of sarcoma. The ultimate goal of this trial is to develop a pan-sarcoma diagnostic tool, and therefore we have include all histologic subtypes. All sarcoma patients in this study were diagnosed with soft tissue sarcoma, also the Ewing patient was a soft tissue Ewing sarcoma. We could further validate these result in the future in other sarcoma types as well including bone sarcoma.
Instead of focusing on the primary or metastatic site of the tumors we think that the histologic subtype and tumor load are more important for the algorithm and therefore we included only this information in the tables of the manuscript.

  1. Regarding patients with active sarcoma, the authors should give precise information about the time of collection. It would be interesting to know if the blood samples for TEPs analysis were collected at the time of diagnosis, pre- or post-operatively or during follow-up. Is there homogeneity when collecting the blood sample in patients with active sarcoma? Samples for training and validation evaluation were collected and processed similarly?

Response: There is indeed heterogeneity in the collection time points of the sarcoma patients, because all patients with active tumor load could be included (at diagnosis, before or during anti-cancer treatment). Despite this, all samples were processed according to a standard operating procedure wet-lab protocol and individuals who handled the samples were blinded for moment of blood collection of the sarcoma patients. Based on the earlier studies of Best et al. with other tumor types including non-small-cell lung cancer and breast cancer, we do expect that all patients with active tumor load would have TEPs with sufficient spliced RNA ‘educated’ signal. We have added statements about the time points of collection and the collection and processing protocol of the blood samples in “Materials and methods, inclusion of patients”, line 262-264 (Blood samples were collected from sarcoma patients with active tumor load before or during anti-cancer treatment (sarcoma series).  All samples were processed according the same standardized protocol of Best et al. [20].). 

3 During the platelet-isolation procedure it seems possible the residual circulating free DNA contamination potentially introduced causes reduced numbers of platelet-RNA sequencing reads. The authors should explain what kind of measures have been implemented to avoid this inconvenience and have false results.

Response: We previously observed significant co-isolation of cell-free DNA together with the platelet RNA (See also Best et al. Cancer Cell 2017). To circumvent this potential cell-free DNA contamination we only selected intron-spanning spliced RNA reads for analysis. This detail has been described in the paragraph “materials and methods, Particle-swarm optimization-enhanced thromboSeq analysis”.

Authors should follow the style of a structured abstract, which is based on the IMRAD structure of a paper.

Response: We do appreciate this comment from the reviewer. Though, we have followed the guidelines of Cancers for a structured abstract, with no headings. We started the abstract with background information to explain the rationale of study and finish the introduction with the aim of the study. Subsequently, we provide a summary of the methods and the most important results. The last sentence highlights the main conclusion of our study. Despite this, if required we are happy to reformat our abstract.

Reviewer 2 Report

In this report, Heinhuis and colleagues developed a self-learning algorithm based on RNA-sequencing of tumor-educated platelets for blood-based sarcoma diagnostics” . First the authors compared expression profiles from 57 sarcoma patients to 103 controls. Then, the authors trained a self-learning support vector machine to subclassify various sarcoma subtypes. While I consider the manuscript interesting and relevant, I am wondering how strong the presented evidence for these key messages is. While results are suggestive, the experimental design does not seem to address specifically the conclusions drawn by the authors. The manuscript lacks detail and could benefit from a more comprehensive explanation of the work.

  • First and foremost, the number of patient samples used for this study is very low, with less than ~10 patients per disease subtype and less than 20 cases for evaluation and validation. Strong conclusions cannot be drawn from such a small cohort. Greater numbers would have strengthened this study, even though the authors highlight it as a proof of principle.
  • The authors state that they identified a unique sarcoma signature from TEP RNA. Yet they did not proof its specificity for sarcoma. How does this profile compare to other cancer?
  • It is not clear for the reader whether the algorithm is based on alternatively spliced RNAs or differentially expressed? Therefore for those who are not familiar with previous work of then group the Thrombo-Seq algorithms should be explained in more detail.
  • In Line 144: the authors report a cut-off 0.5. What does this refer to, is it the TEP score that is introduced later on? The explanation of the TEP score in lines 166/167 is not really informative.
  • How does the TEP score correlate with the stage of the disease? Although more than two thirds of the patient had metastatic disease, it would be interesting to see if metastatic patients show higher scores? What was the stage of the false negative cases?
  • Eight out of eleven false positive controls were former sarcoma patients, but none of them recurred. Therefore the statement of be a promising technique for recurrence monitoring is overstated.
  • Figure 1 is not informative at all, one could include the correctly and falsely classified patients in the Figure.

Reviewer 3 Report

The present manuscript shows a proof of concept study to assess whether tumor-educated platelets (TEMs) can be used as blood-based biomarkers for sarcoma patients. The authors analyzed a cohort of 57 sarcoma patients, and a group of 103 control samples divided into 65 healthy controls and 38 former sarcoma patients without active disease for more than 3 years. The TEPs RNA profiles of the two biological groups were compared and a considerable amount of TEPs RNAs differentially expressed were identified, demonstrating their altered expression in sarcoma patients. The authors applied a self-learning classification algorithm to select a panel of biomarker that could discriminate sarcoma patients from healthy controls. The analysis was performed by dividing the available dataset into training, evaluation and validation series. The accuracy of the classificator was high in both evaluation and validation series, showing that a specific TEP RNA signature can be associated to sarcoma patients. The authors conclude that the identified TEP RNA analysis can be effectively applied as blood-derived tool for sarcoma diagnosis. 

The present pilot study lays the groundwork for the identification of blood-based biomarker for sarcoma patients. The current absence of blood molecular marker for early diagnosis in such disease makes the aim of the project innovative and useful from a clinical point of view. Despite the small sample size, the statistical analysis pointed out the presence of a TEP signature that could be the starting point for more accurate and reliable evaluations. There are a few points should be clarified:

  1. The results of the first differential expression analysis is reported in supplementary table 1. The following optimization of the results led to the identification of a panel of 884 transcripts representing the sarcoma signature of TEPs. It would be better to report these 884 genes in a table. Moreover, authors could discuss the relevance of the most differentially expressed transcripts in relation to the disease. Are there any differentially expressed genes that should be highlighted? 
  2. It is not clear how the TEP-score has been generated. Could the authors provide a more detailed description in “Material and Methods” section?
  3. In the discussion, authors claimed that they showed that thromboSeq can be a promising technique for monitoring sarcoma recurrence (page 11 line 198, page 12 line 199). Actually, this statement is not fully supported by the results. This could be demonstrated only if a correlation analysis is performed, by considering the information of relapse occurrence for the analyzed patients. In this case the authors suppose that the same analysis could be applied to detect a TEP signature predicting relapse occurrence, but it is not demonstrated and, thus, this concept should not be expressed so firmly. 

Author Response

Comments of reviewer #3: 1. The results of the first differential expression analysis is reported in supplementary table 1. The following optimization of the results led to the identification of a panel of 884 transcripts representing the sarcoma signature of TEPs. It would be better to report these 884 genes in a table. Moreover, authors could discuss the relevance of the most differentially expressed transcripts in relation to the disease. Are there any differentially expressed genes that should be highlighted? 

Response: We are thankful for pointing out this issue. The levels reported in Table S1 are based on the full dataset consisting of all 160 samples (see also Figure 1a). However, the panel of 884 RNA transcripts identified to allow the most optimal prediction of the samples included in the evaluation set, is based on the samples in the training set only (n=55). We now added Table S2 showing the names of the 884 identified platelet transcripts used for SVM-algorithm training and validation. Furthermore, we have added three columns indicating whether a transcript was identified in other TEP classifiers as well (Best et al. Cancer Cell 2017 (non-small-cell lung cancer (NSCLC) versus non-cancer controls) and Best et al. Nature Protocols 2019 (lower-grade glioma (LGG) versus controls) or are unique for this classifier (see also Figure S3; venn-diagram analysis). Furthermore, gene ontology and analysis of individual transcripts would indeed be of importance to understand functionality. However, we will need more time to comprehensively study the gene patterns, which is complicated by the absence of Gene Ontology programs for anucleated cells. The pathways derived from current programs need careful curation and analysis since they were developed on expression profiles of nucleated cells.

  1. It is not clear how the TEP-score has been generated. Could the authors provide a more detailed description in “Material and Methods” section?

Response: We provided more information about the selection of the TEP score to line 185-193, “The cut-off value was based on the most accurate fit with a high sensitivity, specificity and accuracy. Because the aim of our study is to evaluate the potential of TEPs in sarcoma diagnostics, the exact cut-off of 0.5 was selected to obtain high specificity to avoid an excess of false-positive samples. The default cutoff of the SVM algorithm (0.5) resulted in a specificity of 88%, sensitivity of 81% and accuracy of 85% for the training series, a specificity of 94%, sensitivity of 84% and accuracy of 80% for the evaluation series, and a specificity of 86%, sensitivity of 88% and accuracy of 87% for the validation series”.

  1. In the discussion, authors claimed that they showed that thromboSeq can be a promising technique for monitoring sarcoma recurrence (page 11 line 198, page 12 line 199). Actually, this statement is not fully supported by the results. This could be demonstrated only if a correlation analysis is performed, by considering the information of relapse occurrence for the analyzed patients. In this case the authors suppose that the same analysis could be applied to detect a TEP signature predicting relapse occurrence, but it is not demonstrated and, thus, this concept should not be expressed so firmly.

Response: We do agree with the reviewer and changed our statement in line 224-226 into “We showed that thromboSeq might be a promising technique in sarcoma diagnostics, however its potential in monitoring of tumor recurrence still needs to be explored”.

Round 2

Reviewer 2 Report

The authors answered most of the reviewer´s questions. The authors performed some additional analyses, which further describe their assay. Regarding the uniqueness of sarcoma signature, it would be interesting to see whether the sole use of the  472 sarcoma-specific would yield the same classification result or outperform the existing algorithm.

Based on the reviewer´s suggestion, the authors explored a possible correlation between stage and TEP-score. Although the hypothesis that metastatic disease results in higher scores could not be confirmed due to the limited number of stage II samples these data are of interest and might be included in the manuscript. This further hints to the limitation of the study and should be discussed.

Author Response

We appreciate the suggestion of the reviewer to further explore the predictive strength of our sarcoma TEP algorithm after exclusion of transcripts that were identified in other TEP classifiers as well. We retrained the algorithm based on the 472 transcripts as suggested by the reviewer and evaluated its predictive power on the evaluation set, the same way the readout for the original 884 transcript classifier was done. We found an AUC of 0.92 on the evaluation set. This AUC seems comparable to the AUC of 0.94 reached by the 884 transcripts signature. However, within the range of all 1000 different algorithm variants generated during the swarm procedure, this algorithm fits within the 5% best-performing algorithms. This is suggesting that the swarm procedure is the most optimal way of classifier development and inclusion of transcripts that are not solely related to sarcoma but other cancers as well contribute to the algorithm performance.
We added a statement in line 162-164, “A variant of our sarcoma classifier trained on the subset of 472 sarcoma specific transcripts as visualized in Figure S2, did not outperform the original swarm-enhanced biomarker panel of 884 transcripts.”
We agree with the suggestion of the reviewer to add a statement about the limited number of included sarcoma patients with low stage disease as a limitation of the study. We added a statement in line 108-109, "We were unable to calculate a correlation between disease-stage and TEP-score because of the limited number of stage II samples (n=2)". And also in the discussion in line 257-258, “Another limitation of this study is that we cannot yet differentiate between the different histologic subtypes and different stages of diseases of sarcoma. To improve this ability of the algorithm, we need to include more patients per histologic subtype and with lower stage of disease in a prospective study.”